# Comparing healthcare systems between the Netherlands and Australia in management for children with acute gastroenteritis

**Anouk A. H. Weghorst**[1]*, **Lena A. Sanci**[2], **Marjolein Y. Berger**[1], **Harriet Hiscock**[3,4], **Danielle E. M. C. Jansen**[1]

1 Department of Primary and Long-Term Care, University of Groningen, University Medical Centre Groningen, Groningen, The Netherlands, 2 Department of General Practice, University of Melbourne, Parkville, Victoria, Australia, 3 Murdoch Children's Research Institute, Health Services Research Group, Melbourne, Victoria, Australia, 4 Department of Paediatrics, University of Melbourne, Parkville, Victoria, Australia

* a.a.h.weghorst@umcg.nl, aahweghorst@gmail.com

## Abstract

### Background

Acute gastroenteritis is a highly contagious disease demanding effective public health and clinical care systems for prevention and early intervention to avoid outbreaks and symptom deterioration. The Netherlands and Australia are both top-performing, high-income countries where general practitioners (GPs) act as healthcare gatekeepers. However, there is a lower annual incidence and per-case costs for childhood gastroenteritis in Australia. Understanding the systems and policies in different countries can lead to improvements in processes and care. Therefore, we aimed to compare public health systems and clinical care for children with acute gastroenteritis in both countries.

### Methods

A cross-country expert study was conducted for the Netherlands and Australia. Using the Health System Performance Assessment framework and discussions within the research group, two questionnaires (public health and clinical care) were developed. Questionnaires were delivered to local experts in the Netherlands and the state of Victoria, Australia. Data synthesis employed a narrative approach with constant comparison.

### Results

In Australia, rotavirus vaccination is implemented in a national program with immunisation requirements and legislation for prevention, which is not the case in the Netherlands. Access to care differs, as Dutch children must visit their regular GP before the hospital, while in Australia, children have multiple options and can go directly to hospital. Funding varies, with the Netherlands providing fully funded healthcare for children, whilst in Australia it depends on which GP (co-payment required or not) and hospital (public or private) they

**Data Availability Statement:** All relevant data are within the paper and/or its Supporting Information files.

**Funding:** This research was supported by the Koninklijke Nederlandse Akademie van Wetenschappen (KNAW) Ter Meulen Grant/Royal Netherlands Academy of Arts and Sciences (KNAW)Medical Sciences Fund, Royal Netherlands Academy of Arts & Sciences (KNAWWF/1085/TMB424). The funding source had no role in the design and conduct of the study; collection, management, analysis, and interpretation of the data; preparation, review or approval of the manuscript; and decision to submit the manuscript for publication.

**Competing interests:** The authors have declared that no competing interests exist.

**Abbreviations:** GP, general practitioner; HSPA, Health System Performance Assessment; WHO, World Health Organization.

visit. Additionally, the guideline-recommended dosage of the antiemetic ondansetron is lower in the Netherlands.

## Conclusions

Healthcare approaches for managing childhood gastroenteritis differ between the Netherlands and Australia. The lower annual incidence and per-case costs for childhood gastroenteritis in Australia cannot solely be explained by the differences in healthcare system functions. Nevertheless, Australia's robust public health system, characterized by legislation for vaccinations and quarantine, and the Netherland's well-established clinical care system, featuring fully funded continuity of care and lower ondansetron dosages, offer opportunities for enhancing healthcare in both countries.

## Introduction

Acute gastroenteritis is a highly contagious disease that leads to significant morbidity, especially among young children [1]. Although the disease is self-limiting, its associated social and economic burdens are substantial [2,3]. For children with acute gastroenteritis, a good public health and clinical care system is required for prevention and early intervention to avoid outbreaks and symptom deterioration [4]. The Netherlands and Australia are both ranked in the top-performing health systems amongst other high-income countries, and both have general practitioners [GPs] as key components of the healthcare system [5,6]. Despite this, there is a lower annual incidence and per-case costs for childhood gastroenteritis in Australia. Variation in these numbers can be a reflection of care pathways, health system funding, structures or service utilizations.

Differences in the annual incidence of acute gastroenteritis episodes per child under five years are evident between the two countries, with 1.96 episodes per child per year in the Netherlands compared to 1.58 episodes per child per year in Australia [7,8]. The incidence rate of a communicable disease can serve as an indicator of the effectiveness of the public health system, encompassing health promotion, vaccination programs, and infectious disease prevention [9]. Besides the variation in incidence, the costs per episode also vary significantly across these countries. The estimated medical costs per episode for children under five years of age in the Netherlands is €55.68 (AUD$ 81.29) compared to €14.37 (AUD$ 20.98) per episode in Australia in 2016 [7,8]. These costs primarily encompass expenses related to GP visits, referrals, and hospitalizations.

Comparative research in these two countries can contribute to healthcare system strengthening by understanding and acknowledging best practices and learning from these best practices [10]. An overview of the healthcare systems in the Netherlands and Australia for children with acute gastroenteritis is lacking. Therefore, this study aimed to better understand the healthcare system of the Netherlands and Australia for children with acute gastroenteritis, focusing on the public health system and the clinical daily care.

## Materials and methods

A cross-country expert study was conducted, among experts from the Netherlands and Australia aiming to compare the public health and clinical daily care for children with acute gastroenteritis. Study methods and findings are reported in accordance with the Consolidated Criteria

for Reporting Qualitative Research [11]. Ethical approval was obtained from the Ethics Committee of the University Medical Centre Groningen (METc 2023/134) and University of Melbourne (2023-26907-39606-3). Informed consent was obtained from participating experts.

## Health system performance assessment framework

The methods of this study were based on procedures used in a prior report comparing health systems between different countries [12]. Study-designed questionnaires were based on the Health System Performance Assessment (HSPA) for Universal Health Coverage framework [13]. The European Observatory on Health Systems and Policies (hosted by the World Health Organization (WHO) Regional Office for Europe) has established the HSPA framework to be able to understand, describe, and compare the functioning of health systems (Fig 1). This framework provides a foundation for policy makers for evaluating health systems by linking their functions to intermediate objectives and final health system goals. Four health system functions have been created to describe the working of the healthcare system: governance; financing; resource generation; and service delivery (see below). Optimizing these functions can improve the intermediate objectives which will lead to better final health system goals [13].

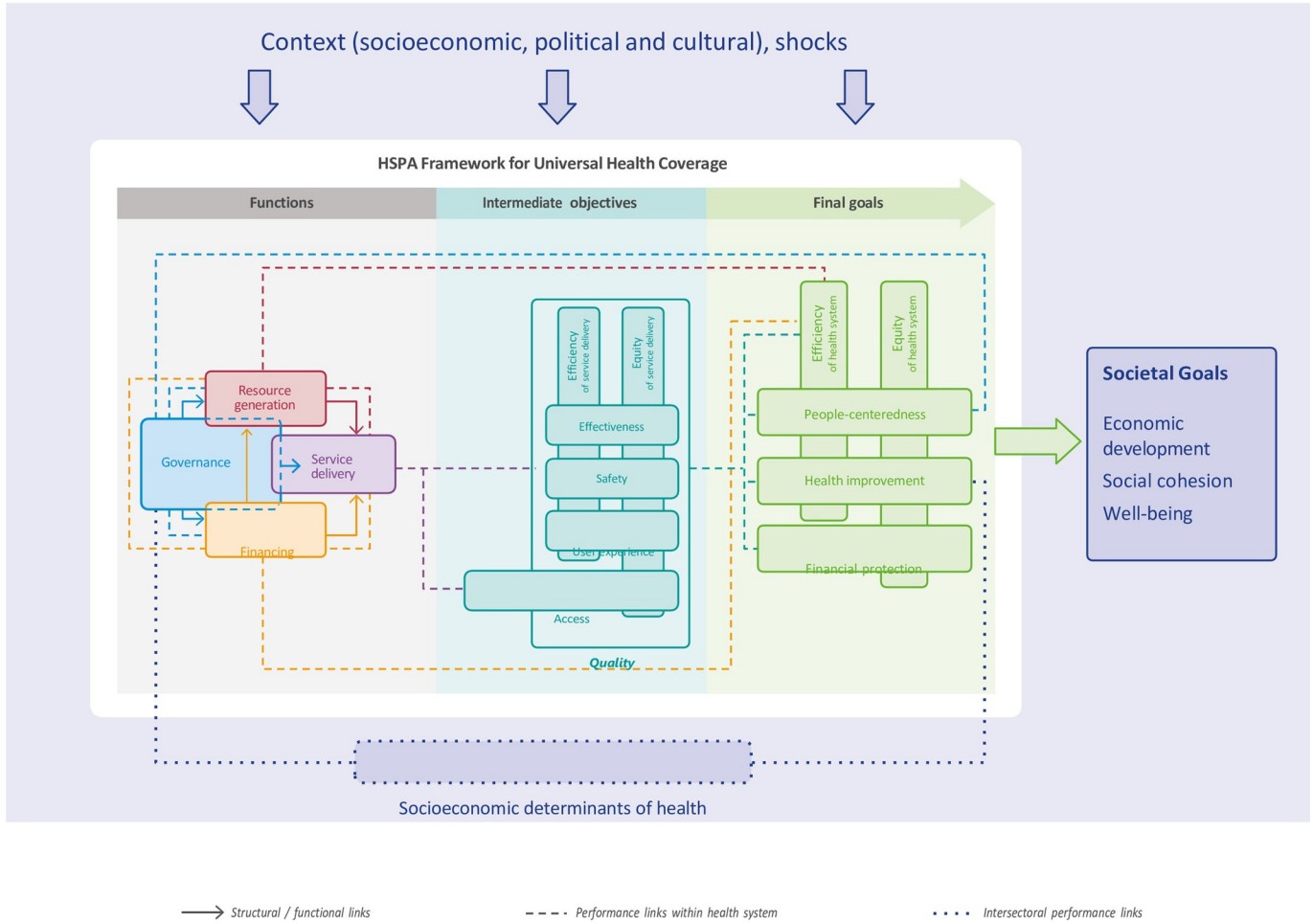

**Fig 1. Health system performance assessment framework–an overview.**

**Governance** is a core health system function, shaping how the other functions are managed and operate. Governance determines the oversight, regulation and policies to effectively address the needs of the country. **Financing** is vital for sustaining the healthcare system by providing the necessary monetary resources for care implementation. **Resource generation** is essential for equipping the healthcare system with essential input, including human resource; infrastructure and medical equipment; and pharmaceuticals and other consumables. **Service delivery** involves the actual provision of medical care, treatment and support to meet the health needs of the individuals. It is a function influenced by the performance of governance, financing and resource generation. It has direct impact on the intermediate objectives, such as access, quality and safety of care.

## Questionnaire

Based on the HSPA framework and discussions within the research group, we developed two questionnaires. One questionnaire focussed on the functioning of the public health system for children with gastroenteritis. The other questionnaire addressed clinical daily care for treating children with gastroenteritis (S1 File). Public health includes the effective outbreak management and prevention of this highly contagious disease. Clinical daily care encompasses the healthcare pathway from the initial onset of gastroenteritis until fully recovered. The questionnaires were piloted among experts and were adjusted if needed. For the experts in the Netherlands, the questionnaires were translated into Dutch.

## Data collection

To collect up-to-date and country-specific information, questionnaires were delivered online to local experts in the Netherlands and the state of Victoria, for the Australian context, in April and May 2023. As these questionnaires aimed to understand the accepted guidelines and regulatory parameters for public health and clinical care system responses only, we considered two experts per category, per country as an adequate sample size. Opinions on the healthcare system were not requested. Public health questionnaires were provided to experts specializing in the public health of infectious diseases. The clinical care questionnaires were distributed among general practitioners, paediatricians and guideline developers. Input from their organizational group or team was welcome.

## Data synthesis

A narrative approach was employed to synthesize the data obtained from the questionnaires, including data from guidelines and other relevant sources provided by the experts. Through the synthesis, a constant comparison was made between data from the Netherlands and Australia. Unless specified, the description of the system for Victoria is the same for all of Australia.

## Results

### Participants

In the Netherlands, two physicians from the National Institute for Public Health and the Environment answered the public health questionnaire. In Australia, an infectious disease physician and a manager responsible for communicable disease prevention and control, both working for the government of Victoria, provided information for the public health questionnaire. The clinical care questionnaire in the Netherlands was completed by a GP, a staff member from the Dutch College of GPs, and a pharmacist. Responses pertaining to clinical care practices in Australia were obtained from a GP specializing in child health and a paediatrician.

The experts, on average, had over 15 years of experience in their professional field. Table 1 gives a summary of the main differences per country.

## Governance

Table 2 gives a comprehensive overview of the government and legislations applicable for children with acute gastroenteritis for both the Netherlands and Australia.

**Public health.** Outbreak management is achieved through the implementation of a systematic approach aimed at rapidly gaining insight into the outbreak. Both countries adopt a multisectoral approach to outbreak management. Acute gastroenteritis is a highly contagious disease with rotavirus being the most common cause [2,14]. In the **Netherlands**, the inclusion of the rotavirus vaccine in the National Immunisation Program is planned for 2024, and participation in the program is voluntary. There is no legislation specifically aimed at prevention. In **Australia**, children are required to stay at home for 48 hours in an outbreak setting. The rotavirus vaccine has been included in the National Immunisation Program in Australia since 1st July 2007. To encourage the number of children fully immunised in line with the National Immunisation Program, the Australian Government initiated two policies. To access family assistance payments, children must meet immunisation requirements under the No Jab No Pay scheme. Children attending childcare in Australia are required to meet the National Immunisation Program under the No Jab No Play legislation 2016, unless there is a medical exception.

**Clinical care.** The **Netherlands** follows a system where patients must be registered with a GP practice to access their services and are not allowed to register with more than one GP. This ensures access to a GP when needed, allows for after-hours primary care, and facilitates the continuity of healthcare and monitoring of health status. In **Australia**, patients are not required to register with a regular GP and have the flexibility to book appointments with their preferred GP, sometimes even consulting multiple GPs on a single day.

## Financing

The financing of the **Dutch** healthcare system is based on social health insurance and managed competition. Dutch citizens are required to obtain health insurance that covers a standard basic benefits package. Insurance premiums are determined by individual insurers. In **Australia**, the healthcare system is financed through Medicare which is the government-funded

**Table 1. Main differences per country.**

|  | The Netherlands | Australia |
|---|---|---|
| Governance | • No legislation aimed at prevention<br>• Rotavirus in the Immunisation Program since 1st of January 2024<br>• GP registration is obligatory, not allowed to register with more than one GP | • No Jab No Pay, No Jab No Play legislation[1]<br>• Rotavirus in the Immunisation Program since 2007<br>• GP registration is not obligatory |
| Financing | • Fully funded care for children under 18 years | • Funding depends on the choice of GP (co-payment required or not) and hospital (public or private) |
| Resource generation | • Ondansetron[2] in syrup (0.1 mg/kg) | • Ondansetron[2] in wafer (8-15kg 2mg, 15-30kg 4mg, >30kg 6-8mg) |
| Service delivery | • GP care always first<br>• Direct visits to emergency department are discouraged | • Multiple options for seeking first care<br>• Direct visits to emergency department are part of organization of daily care |

[1]No Jab No Pay: To access family assistance payments, immunisation is required. No Jab No Play: To attend childcare, immunisation is required. [2] an antiemetic medication.

**Table 2. Government and legislations.**

| Netherlands | | Australia | |
|---|---|---|---|
| Ministry of Health, Welfare and Sport (VWS) | Government department responsible for public health, welfare, and sports policy. | Department of Health | Government department responsible for the administration and oversight of healthcare, public health and related services. |
| Health and Youth Care Inspectorate (IGJ) | Governmental agency responsible for monitoring and regulating healthcare and youth care. | Safer Care Victoria | Governmental agency responsible for driving improvements in the quality and safety of healthcare services. |
| Dutch Medical Treatment Contracts Act (WGBO) | Legislation that governs the relationship between healthcare professionals and patients. | Health Services Act | Legislation that governs various aspects of healthcare services and facilities. |
| Healthcare Professionals Act (Wet BIG) | Legislation that regulates the practice of healthcare professionals. | Health Practitioner Regulation National Law Act | Legislation that governs the registration, regulation, and professional conduct of healthcare professionals. |
| Healthcare Insurance Act (Zvw) | Legislation that governs the mandatory health insurance system. | Health Insurance Act (Australia) | Legislation that establishes the legal framework for the country's public health insurance system known as Medicare. |
| General Data Protection Regulation | Data protections and privacy regulation implemented by the European Union. | Privacy and Data Protection Act | Legislation that governs the protection of personal information, including health-related data. |
| Medicines Act | Legislation that regulates the production, distribution, sale, and use of medicinal products. | Pharmacy Regulation Act | Legislation that regulates the practice of pharmacy. |
| Public Health Act (Wpg) | Legislation that governs the public health policy and public health interventions. | Public Health and Wellbeing Act | Legislation that establishes the framework for public health and wellbeing measures. |

healthcare system in Australia. Medicare is accessible to all Australian citizens. Citizens can choose to purchase extra private health insurance to access additional healthcare services (largely hospital care) not covered by Medicare.

**Public health.** Both countries fully cover the expenses associated with rotavirus vaccinations through their Public Health Services.

**Clinical care.** The funding system for GPs in the **Netherlands** involves a combination of fee-for-service and capitation-based reimbursement, along with some additional payments for specific services. The fee is based on the type of service and is subject to negotiation between healthcare providers and health insurers. The capitation fee is determined based on factors like the patient's age, gender, and health status. Fees are adjusted annually to account for inflation and changes in the practice's patient demographics. The Netherlands has a mandatory health insurance system, and individuals are required to have basic health insurance coverage. Dutch children under 18 years are automatically covered by their caregivers' insurance and clinical care for children with acute gastroenteritis is fully funded through government contribution from taxes, meaning no out-of-pocket costs for patients. This includes prescription of medication (eg. The anti-emetic ondansetron). Over-the-counter medications (eg. oral rehydration solution, paracetamol) are paid for by caregivers. Hospital care for children with acute gastroenteritis in the Netherlands is also fully covered.

In **Australia**, the payment structure for GPs is primarily based on a fee-for-service model. GPs charge a fee for each service provided to patients. Patients typically pay the GP directly and then claim a rebate from Medicare. Medicare does not remunerate the GP but it reimburses the patient for services provided. However, the remuneration has not been adjusted for inflation and rising costs for a decade which has been associated with increased co-payments (ie., out-of-pocket costs) for patients. Some GPs offer 'bulk-billing', a term which means they do not charge the patient a co-payment; Medicare covers the full consultation cost, and patients authorise Medicare to pay the GP directly on their behalf.

The costs of primary care for caregivers of Australian children with acute gastroenteritis depends on the choice of GP (co-payment or not). Medication for childhood gastroenteritis prescribed in general practice is typically issued as a private prescription, and the caregivers are responsible for the costs. For hospital care, it depends on whether caregivers choose public or private hospitals. Care provided in public emergency departments and public hospitals is fully covered by state and federal governments. Medication prescribed in the public hospital is included in the hospital visit. For private hospital care, state and federal governments cover 75% of the hospital and medical fees. The remaining fees are billed to the caregivers, and depending on their private health insurance, certain fees might be covered.

## Resource generation

**Health workforce.** Organizations and professionals involved in the public health workforce of infectious gastroenteritis outbreaks in children in both countries include: the institution where the outbreak occurred (i.e., schools and child day-care centres), public health services (infectious disease control doctors and nurses, infection prevention experts, youth health care doctors), laboratories (medical microbiology doctors), GPs, and paediatricians. In both countries, management of acute gastroenteritis, particularly dehydration in children, is covered in medical school and training for GPs and paediatricians.

**Infrastructure and medical equipment.** For public health, both countries offer stool testing to identify the infectious agent causing the gastroenteritis outbreak. For clinical care, the availability of medical equipment for the management of childhood gastroenteritis in primary care is minimal in both countries. Primary care is not typically set up to monitor vital signs in an ongoing way or give fluids other than orally. Emergency departments and paediatricians in-hospital in both countries have access to a wide range of additional diagnostics, including point-of-care blood testing.

**Pharmaceuticals and other consumables.** For public health, both countries offer access to the rotavirus vaccine. However, in the **Netherlands** the rotavirus vaccine is not included in the National Immunisation Program whereas in **Australia** it is included. For clinical care in both countries, over-the-counter measures (i.e., paracetamol, ibuprofen/naproxen, oral rehydration solutions) are available through pharmacies, drugstores or supermarkets. Prescribed ondansetron, an antiemetic, is available through pharmacies in syrup (Netherlands) or wafer form (Australia).

## Service delivery

**Public health.** For the prevention of rotavirus gastroenteritis, in **Australia** it is recommended to receive the first vaccine dose by 14 weeks of age followed by a second dose by 24 weeks of age.

Both countries have established national guidelines for the public health response to managing infectious gastroenteritis outbreaks in children which outline a step-by-step plan [15,16]: surveillance and detection; reporting to Public Health Service; investigation and epidemiological analysis; control measures; communication and education strategies; and follow-up and evaluation.

**Clinical care.** In both countries, access to clinical daily care for children with acute gastroenteritis is initiated by caregivers. During working hours, the first point of contact is typically the GP. In the **Netherlands**, the care for children with acute gastroenteritis emphasizes initial contact with the regular GP during working hours This GP has knowledge of the child's medical history. The GP can provide guidance or make referrals to emergency care if necessary. After working hours, caregivers can access unscheduled care through GP out-of-hours

facilities operated by larger cooperatives of GPs. At the out-of-hours facilities, locum GPs are available for (telephone) consultations. Every citizen has access to this after-hours care. The Dutch healthcare system discourages direct and unscheduled visits to the emergency department without a GP referral, highlighting a preference for a more formalized route through primary care. In **Australia**, while 80% of patients are said to have a regular GP [17], caregivers have multiple avenues for seeking care, including booking an appointment with any GP, contacting a telephone nurse for basic advice, scheduling virtual emergency department consultations through telehealth services, or visiting the emergency department to see a clinician. After working hours, access to GP services may be limited and clinical care can be provided by locum GPs or through the options mentioned earlier. The Australian system acknowledges the various ways in which unscheduled care may be accessed, providing caregivers with options beyond the GP visit, especially after working hours.

Guidelines for GPs and paediatricians are available online in both countries [18–21]. These guidelines cover acute gastroenteritis background, assessment and management.

In both countries, the primary recommendation for the clinical care of children with acute gastroenteritis is to prioritize rehydration as the initial treatment approach, primarily through oral rehydration solutions. The use of antibiotics and anti-diarrheal medications are not recommended for the treatment of children with acute viral gastroenteritis in both countries. As of December 2022, the **Netherlands** introduced a recommendation for a single dose of oral ondansetron syrup (0.1 mg/kg), an antiemetic medicine, for primary care management of gastroenteritis, whereas it previously was only advised in secondary care provided by paediatricians. In **Australia**, ondansetron is recommended in a higher weight base dose (8–15 kg 2mg; 15-30kg 4mg; >30kg 6-8mg) in the form of a wafer. After triage in emergency departments, there is early access to oral rehydration and ondansetron. Hospital management by a paediatrician is in both countries based on the severity of dehydration (mild, moderate or severe). For children with mild to moderate dehydration enteral rehydration is preferred. Intravenous dehydration is recommended for severely dehydrated children or children who cannot tolerate enteral rehydration [20,21].

Both countries offer online information for caregivers of children with acute gastroenteritis encompassing information about aetiology, symptoms, treatment advice, when to seek medical assistance, and preventive measures [22,23]. In the **Dutch** resource, written information is supported with a video. **Australian** guidelines recommend that children should not refrain from eating for more than 24 hours, while Dutch guidelines state that a few days without or with reduced food intake does not significantly affect the child.

## Discussion

A comparative synthesis of healthcare systems of two top-performing, high-income countries, the Netherlands and Australia, with the focus on public health and clinical daily care for children with acute gastroenteritis was performed. We wanted to reflect on the differences in incidence and costs between two countries with high standard healthcare. We recognize that these differences are caused by multiple factors in almost all levels of care. One should be aware of this complexity. Based on our results each country can reflect on the differences and evaluate whether adaptation would be feasible and effective in their own setting.

### Public health

While the Netherlands and Australia have similar goals and step-by-step outbreak management plans aiming to promptly address outbreaks, they diverge in their strategies regarding vaccination and legislation for disease prevention. Rotavirus is the most common cause of

severe gastroenteritis in young children and is a primary pathogen among hospitalized children with gastroenteritis [2,14]. In Australia, the introduction of a free rotavirus vaccine into the National Immunisation Program in 2007 resulted in a significant reduction in rotavirus-positive tests [24]. Moreover, the hospital admission rate showed a 62% reduction after the free rotavirus vaccine was implemented in Australia [25]. With the implementation of the 'No Jab No Pay' and 'No Jab No Play' legislations, an increase in full vaccination coverage among children in Australia was seen [26]. In contrast, the Netherlands does not yet include the rotavirus vaccine in its National Immunisation Program and lacks legislation restricting non-vaccinated children. It is plausible to hypothesize that effective immunisation and improved adherence to the immunisation program in Australia, results in less severe rotavirus cases, potentially leading to fewer hospital admissions and reduced healthcare costs. One might assume that similar legislation for immunisation will have the same flow-on benefits in the Netherlands, but the question of whether this type of legislation would be tolerated by Dutch society needs exploring.

## Clinical care

In both countries, the primary goal in managing childhood gastroenteritis is rehydration. Oral rehydration solutions are recommended, while antibiotics and anti-diarrheal medications are discouraged, aligning with international guidelines [27]. In the Netherlands, the recommended single dose of oral ondansetron is 0.1 mg/kg, while Australia advises a higher single dosage regimen [8-15kg 2mg; 15-30kg 4mg; >30kg 6-8mg] [19,28] consistent with previous research [29,30]. The lower dosage strategy in the Netherlands is based on a more recent randomized controlled trial that found [cost-]effectiveness at a lower dosage [31,32]. Furthermore, another study has revealed that children with acute gastroenteritis who received higher doses of ondansetron did not experience a greater reduction in vomiting, nor did they require less intravenous rehydration or hospitalizations compared to children who received lower doses [33]. As there seems no added benefit for higher single doses of oral ondansetron and emphasizing the importance of minimizing the risk of side effects, it could be advisable for Australia to consider adopting a lower single dose of ondansetron in their clinical guidelines.

## Continuity of care

Effective management of childhood gastroenteritis requires safety netting advice, including dehydration and alarm symptom recognition, along with guidance on help-seeking [34]. The quality of safety netting relies on the GP-patient relationship, and a lack of care continuity hampers its provision [34,35]. Research also highlights the benefits of maintaining continuity of care in general practice and accessing the preferred GP can reduce emergency admissions [36]. Additionally, gatekeeping practices are associated with reduced healthcare utilization and the likelihood of fewer hospitalizations [37]. In the Netherlands, the predominant pathway for children with gastroenteritis involves initially consulting their familiar, fully funded GP before entering the hospital. However, gastroenteritis ranks among the top five diagnoses for children seeking out-of-hours primary care centers in the Netherlands, where multiple GPs work in shifts to provide care outside regular working hours [38]. In Australia, although it is reported that 80% of the patients have a regular GP [17], this is not obligatory and parents have diverse care-seeking options. Whilst our study cannot directly quantify the impacts of continuity of care in prevention or management of childhood gastroenteritis, based on previous research, both countries should be aware of optimizing care continuity, focusing on the establishing GP-patient relationships, as this could affect the actual care delivery for children with gastroenteritis.

### Strengths and limitations

Strengths of this study were that we used the HSPA framework established by the WHO, which gave us a full understanding of the health system functioning in both countries and we placed emphasis on the public health as well as clinical daily care. Moreover, the research team responsible for the formulation and evaluation of the questionnaire comprised researchers from both participating countries, thereby enriching the depth of knowledge and expertise applied in the study. Nonetheless, a limitation of this study could be that we only surveyed two experts per category per country. We decided this was an adequate sample size as we aimed to understand the National published guidelines and regulatory parameters and opinions were not requested. We selected experts who possessed considerable experience (on average >15 years) in the field of public health or clinical care and input of their organization was welcome. Lastly, it is worth noting that we used Victoria for the Australian context. However, the measures described here are national and not varying by state in Australia. In addition, the primary care system and National Immunisation program funding and policies are run by the Commonwealth and do not vary in application by jurisdiction.

### Conclusions

Healthcare approaches for organizing and providing healthcare for children with acute gastroenteritis varies between the Netherlands and Australia. The lower annual incidence and per-case costs for childhood gastroenteritis in Australia cannot solely be explained by the differences in healthcare system functions. Nevertheless, Australia's robust public health system, characterized by legislation for vaccination and quarantine, and the Netherland's well-established clinical care system, featuring fully funded continuity of care and lower ondansetron dosages, offer opportunities for enhancing healthcare in both countries.

### Supporting information

**S1 File. Questionnaires.**
(DOCX)

**S2 File. Inclusivity in gloval research questionnaire.**
(DOCX)

### Acknowledgments

We would like to thank Dheepa Rajan and Katja Rohrer for their guidance on the Health System Performance Assessment Framework. Our sincere thanks go to all the experts who participated in this research, generously contributing their time and expertise.

### Author Contributions

**Conceptualization:** Anouk A. H. Weghorst, Lena A. Sanci, Marjolein Y. Berger, Harriet Hiscock, Danielle E. M. C. Jansen.

**Formal analysis:** Anouk A. H. Weghorst, Danielle E. M. C. Jansen.

**Methodology:** Anouk A. H. Weghorst.

**Supervision:** Lena A. Sanci, Marjolein Y. Berger, Harriet Hiscock, Danielle E. M. C. Jansen.

**Writing – original draft:** Anouk A. H. Weghorst.

**Writing – review & editing:** Lena A. Sanci, Marjolein Y. Berger, Harriet Hiscock, Danielle E. M. C. Jansen.

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
