## [Decision Letter · Decision Letter 0]

15 Feb 2024

PONE-D-23-40570Comparing healthcare systems between the Netherlands and Australia in management for children with acute gastroenteritisPLOS ONE

Dear Dr. Weghorst,

Thank you for submitting your manuscript to PLOS ONE. After careful consideration, we feel that it has merit but does not fully meet PLOS ONE’s publication criteria as it currently stands. Therefore, we invite you to submit a revised version of the manuscript that addresses the points raised during the review process.

A rebuttal letter that responds to each point raised by the academic editor and reviewer(s). You should upload this letter as a separate file labeled 'Response to Reviewers'.• A marked-up copy of your manuscript that highlights changes made to the original version. You should upload this as a separate file labeled 'Revised Manuscript with Track Changes'.• An unmarked version of your revised paper without tracked changes. You should upload this as a separate file labeled 'Manuscript'.

We look forward to receiving your revised manuscript.

Kind regards,

Victor Daniel Miron

Academic Editor

PLOS ONE

Journal Requirements:

3. Please expand the acronym “KNAW” (as indicated in your financial disclosure) so that it states the name of your funders in full.

4. We note that your Data Availability Statement is currently as follows: "All relevant data are within the manuscript and its Supporting Information files."

**Reviewers' comments**:

Reviewer's Responses to Questions

**Comments to the Author**

1. Is the manuscript technically sound, and do the data support the conclusions?

Reviewer #1: Partly

Reviewer #2: Partly

2. Has the statistical analysis been performed appropriately and rigorously? 

Reviewer #1: N/A

Reviewer #2: N/A

3. Have the authors made all data underlying the findings in their manuscript fully available?

Reviewer #1: Yes

Reviewer #2: Yes

4. Is the manuscript presented in an intelligible fashion and written in standard English?

Reviewer #1: Yes

Reviewer #2: No

5. Review Comments to the Author

Reviewer #1: Dear Dr Weghorst,

This is an interesting and unique manuscript. In the data collection is there a reference that one can use to justify the sample size that your team selected?

I find it odd that a significant section of the unscheduled healthcare system has been omitted in this analysis namely the emergency department (ED) . When one studies the structures involved in healthcare one of the more sizeable elements is the unscheduled care delivered in an ED. In fact, many would view in certain jurisdictions i.e Australia that the role played by the ED is significant as parents/patients often attend directly to the ED without referral from primary care. Most ED attendances result in a discharge home after an episode of ED / ED guideline directed care ( in some countries for every 7 children with AGE; 6 go home 1 is admitted under a paediatrician). The care delivered on this setting is not just by paediatricians but more likely to be by emergency medicine consultants and staff. I'm aware this may not be the same in the Netherlands as a Paediatric emergency medicine specific training scheme does not exist in that country ( it does in Australia). This omission is obvious once stated but seems to be skirted around by the manuscript. It is a weakness of the manuscript and should be openly declared.

In page 13 we learn that that for children to attend an ED directly is not customary but this needs greater quantification . What %-age are direct attendances in both jurisdictions ? I suspect there will be another difference here that needs to be mentioned.

Is there a reference or a reasoning behind the statement that what happens in Victoria is representative of all Australia?

I see no mention of public health resources aimed directly at families at home. I suspect that would fall under resource generation.

Congratulation on this piece of work. I believe it is unique and additive to the literature. Please consider the critique and incorporate it as you see is reasonable in the next version. I look forward to your responses

Reviewer #2: Thank you for the opportunity to review this interesting paper, which applies a WHO policy analysis framework to compare the health systems of two countries with respect to managing and preventing gastroenteritis in children.

There are two main considerations that I would recommend be addressed to strengthen the manuscript.

Firstly, it would help the reader to make it clearer the rationale for undertaking this comparative analysis and what the authors aim to understand by doing so. The authors state that they want to understand and acknowledge best practices, but how do we know what a best practice is? In the abstract, it is not until the conclusion that the reader learns what the main differences in outcomes are between the two counties. This gives the rationale for the study and why the how the different functions (governance, financing etc) of the health system operate in each setting that may be relevant and illuminating for policy.

Secondly with respect to the methodology, the framework used worked well to break down the components of each health system to facilitate comparison. However, it would help the reader to be more explicit in how it was applied for this specific problem of gastroenteritis in children. Two participants for each major health system component (public health and clinical services) for each county seems a very small sample. Is there literature to support that this is sufficient?

Greater clarity on what the authors are hoping to explain and how the results address this would improve the manuscript.

Some additional comments.

Line 58 It would help to directly refer to what you mean by ‘functions’, as not all readers may be familiar with the WHO HSPA. What the authors mean by service delivery as defined in the methods needs clarification.

Public health is qualified as outbreak management several times, but its seems that broader public health functions are considered (outbreak management AND disease control through immunisation). Could the authors please clarify how they are defining this?

Re table 1, could you summarise the key differences between the policy and regulatory settings that may explain the problem you are interested in (presumably why Australia has lower incidence and lower cost)? Some of the sections in the results seemed to switch between ideas, particularly the one on clinical care under financing. Review of this would improve readability.

Re registration with a single provider, how does this ensure access when needed as well as after-hours care? Further explanation for audiences that are unfamiliar with the Dutch system would be helpful.

Re financing in Australia and PHI. My understanding it that PHI covers care provided as a private hospital patient (in a public or private hospital) as well as dental, psychological and allied health care but not medical services or medicines.

The authors mention outbreak management as a key public health function of both systems. Is this funded differently?

The authors may like to consider using standard terminology for describing the funding arrangements for clinical services (eg fee for services, capitation etc) as this might be more readily understood by international audiences. Re funding in Australia, Medicare does not remunerate the GP, it reimburses the patient for services provided. This has not been adjusted for 10 + years resulting in increased co-payments and out of pocket costs for the patient.

Why does differences in use of Ondansetron matter for your research question? Re the PBS, this does not cover paracetamol or Oral rehydration solutions (or ondansetron generally) except for very specific populations or conditions (ie. palliative care for ondansetron).

Under resource generation: It would be helpful if the authors could explain how the public health response to outbreaks differs (if it does) between the countries; this may also account for differences in incidence.

Line 216 what do the authors mean by medical equipment? Are there many differences between community vs hospital based pathology services (that are relevant to childhood GE)?

A summary table of the key differences, with emphasis on those that are expected to contribute to lower incidence and costs, would be very helpful to the reader.

Sections of the results are repeated in the discussion, and possibly some of what is in the discussion would be better placed in the results and should be reviewed. The relevance of ondansetron use to the study aim needs further explanation. The logic of the argument in the section on continuity of care was unclear to me. While continuity is a good thing for a multitude of reasons, how had your study shown that continuity of care contributed to managing and preventing GE in childhood?

6. PLOS authors have the option to publish the peer review history of their article (what does this mean?). If published, this will include your full peer review and any attached files.

Reviewer #1: **Yes: **Michael Barrett

Reviewer #2: No

---

## [Author Response · Author response to Decision Letter 0]

7 Apr 2024

We would like to thank the editor for the opportunity to submit a revised version of our manuscript (PONE-D-23-40570). We thank the reviewers for the constructive comments and suggestions to improve our manuscript. The detailed response to each of their comments is provided in this document. We hope our revisions will meet the expectations put forward by the reviewers. The page and line numbers we refer to correspond to the ‘track changes’ document. 

Reviewer 1: 

1. This is an interesting and unique manuscript. In the data collection is there a reference that one can use to justify the sample size that your team selected?

Thank you for your comment. The methods of this study were based on a previous performed advisory report comparing public health systems between Denmark, England, Italy, Latvia, the Netherlands, Australia, British Colombia (Canada) and Singapore. In that report, to collect up-to-date and country-specific information, two national experts in public health were approached for each country to gather information regarding the building block model. We replicated this approach.

We therefore added ‘The methods of this study were based on a previous performed advisory report comparing health systems between different countries (Ref 12 (added))’ (Page 6, line 93-94). 

Moreover, we were not interested in opinions about the healthcare systems, but collected actual objective data. We selected experts with experience on average >15 years and encouraged them to ask for input of their organization if they couldn’t answer a question input of their organization was welcome. With two experts per category per country we received the information we needed about the healthcare systems. 

2. I find it odd that a significant section of the unscheduled healthcare system has been omitted in this analysis namely the emergency department (ED). When one studies the structures involved in healthcare one of the more sizeable elements is the unscheduled care delivered in an ED. In fact, many would view in certain jurisdictions i.e. Australia that the role played by the ED is significant as parents/patients often attend directly to the ED without referral from primary care. Most ED attendances result in a discharge home after an episode of ED / ED guideline directed care ( in some countries for every 7 children with AGE; 6 go home 1 is admitted under a paediatrician). The care delivered on this setting is not just by paediatricians but more likely to be by emergency medicine consultants and staff. I’m aware this may not be the same in the Netherlands as a Paediatric emergency medicine specific training scheme does not exist in that country ( it does in Australia). This omission is obvious once stated but seems to be skirted around by the manuscript. It is a weakness of the manuscript and should be openly declared.

Thank you for the comment. We agree with the reviewer that the unscheduled care was not highlighted enough and we therefore changed the original section to 

‘In the Netherlands, the care for children with acute gastroenteritis emphasizes initial contact with the regular GP during working hours. This GP has knowledge of the child’s medical history. The GP can provide guidance or make referrals to emergency care if necessary. After working hours, caregivers can access unscheduled care through GP out-of-hours facilities operated by larger cooperatives of GPs. At the out-of-hours facilities, locum GPs are after working hours available for (telephone) consultations. Every citizen can access this after-hours care. The Dutch healthcare system discourages direct and unscheduled visits to the emergency department without a GP referral, highlighting a preference for a more formalized route through primary care. In Australia, while 80% of patients are said to have a regular GP (15), caregivers have multiple avenues for seeking care, including booking an appointment with any GP, contacting a telephone nurse for basic advice, scheduling virtual emergency department consultations through telehealth services, or visiting the emergency department to see a clinician. After working hours, access to GP services may be limited and clinical care can be provided by locum GPs or through the options mentioned earlier. The Australian system acknowledges the various ways in which unscheduled care may be accessed, providing caregivers with options beyond the traditional GP visit, especially after working hours.’ (now on pages 14 and 15) (in bold the most important differences) 

 To also include the unscheduled care. 

3. In page 13 we learn that that for children to attend an ED directly is not customary but this needs greater quantification . What %-age are direct attendances in both jurisdictions ? I suspect there will be another difference here that needs to be mentioned.

We reformulated the sentence as per above and now state that the Dutch system discourages direct ED attendance. In this manuscript we do not provide exact numbers as that was not the purpose of our research. 

4. Is there a reference or a reasoning behind the statement that what happens in Victoria is representative of all Australia?

At first, we wanted to compare the state of Victoria with the Netherlands because of practical reasons. We wanted to have the clinical experts from the same state. After further research, we found that the same public health and clinical care management broadly applies to all the states of Australia. This is because there is the same approach to funding (i.e. predominantly state government for EDs and federal government for primary care) across Australia. We therefore took Australia as a whole.

5. I see no mention of public health resources aimed directly at families at home. I suspect that would fall under resource generation.

Thank you for the comment. Access to public health resources for families dealing with children with acute gastroenteritis is facilitated by the health authorities. On page 16 we stated the information parents can find at home. We also stated differences between the Australian and Dutch guidelines for parents at home (Australian guidelines recommend that children should not refrain from eating for more than 24 hours, while Dutch guidelines state that a few days without or with reduced food intake does not significantly affect the child.). 

Moreover, on page 14 we stated what pharmaceuticals and other consumables are available over-the-counter. 

Congratulation on this piece of work. I believe it is unique and additive to the literature. Please consider the critique and incorporate it as you see is reasonable in the next version. I look forward to your responses.

Thank you very much! 

Reviewer 2:

Thank you for the opportunity to review this interesting paper, which applies a WHO policy analysis framework to compare the health systems of two countries with respect to managing and preventing gastroenteritis in children.

There are two main considerations that I would recommend be addressed to strengthen the manuscript.

1. Firstly, it would help the reader to make it clearer the rationale for undertaking this comparative analysis and what the authors aim to understand by doing so. The authors state that they want to understand and acknowledge best practices, but how do we know what a best practice is? In the abstract, it is not until the conclusion that the reader learns what the main differences in outcomes are between the two counties. This gives the rationale for the study and why the how the different functions (governance, financing etc) of the health system operate in each setting that may be relevant and illuminating for policy.

Thank you for the comment. We have now tried to make the rationale for undertaking this analysis clearer by adding the sentences in the Abstract (Page 2, line 25-28) 

‘However, there is a lower annual incidence and per-case costs for childhood gastroenteritis in Australia. Understanding the systems and policies in different countries can lead to improvements in processes and care. Therefore, in this study we aimed to compare public health and clinical care for children with acute gastroenteritis in both countries.’

and in the Introduction (Page 4, line 61-64 and line 77-80): 

‘…there is a lower annual incidence and per-case costs for childhood gastroenteritis in Australia. Variation in these numbers can be a reflection of care pathways, health system funding, structures or service utilizations.’ ……. ‘An overview of the healthcare systems in the Netherlands and Australia for children with acute gastroenteritis is lacking. Therefore, this study aimed to better understand the healthcare system of the Netherlands and Australia for children with acute gastroenteritis, focusing on the public health system and the clinical daily care.’ 

2. Secondly with respect to the methodology, the framework used worked well to break down the components of each health system to facilitate comparison. However, it would help the reader to be more explicit in how it was applied for this specific problem of gastroenteritis in children. Two participants for each major health system component (public health and clinical services) for each county seems a very small sample. Is there literature to support that this is sufficient?

Thank you for the comment. In the first question of the first reviewer, we answer this question. 

Greater clarity on what the authors are hoping to explain and how the results address this would improve the manuscript.

Some additional comments: 

3. Line 58 It would help to directly refer to what you mean by ‘functions’, as not all readers may be familiar with the WHO HSPA. What the authors mean by service delivery as defined in the methods needs clarification.

We changed the sentence (see question 1 reviewer 2) and ‘functions’ is no longer in that sentence. We use the word ‘functions’ on page 6 line 100, 101, 103. Functions of the healthcare systems is an overarching term covering the governance, financing, resource generation, and service delivery. We hope that by adding the figure directly under the text, it will make it clear what the meaning of functions is. 

To clarify what service delivery means, we added the sentence:

 ‘Service delivery involves the actual provision of medical care, treatment and support to meet the health needs of the individuals.’ (Page 7, line 112-113).

4. Public health is qualified as outbreak management several times, but its seems that broader public health functions are considered (outbreak management AND disease control through immunisation). Could the authors please clarify how they are defining this?

We agree that the public health is broader than outbreak management, as we also mention the immunizations and legislations. We therefore removed ‘(outbreak management)’ if it was stated after public health. (Page 6, line 86; Page 7, line 132; Page 18, line 320; Page 20, line 386)

5. Re table 1, could you summarise the key differences between the policy and regulatory settings that may explain the problem you are interested in (presumably why Australia has lower incidence and lower cost)? Some of the sections in the results seemed to switch between ideas, particularly the one on clinical care under financing. Review of this would improve readability.

Thank you for your comment. We agree that some sections in the results switched between the Netherlands and Australia and reformulated it to first mentioning the Dutch healthcare system, followed by the Australian. 

We also added the table below to give a comprehensive overview of the main differences per country. (Page 9, Line 154)

Table 1. Main differences per country

 The Netherlands Australia

Governance • No legislations aimed at prevention

• Rotavirus in the Immunization Program since 1st of January 2024

• GP registration is obligatory, not allowed to register with more than one GP • No Jab No Pay, No Jab No Play legislations

• Rotavirus in the Immunization Program since 2007

• GP registration is not obligatory

Financing • Fully funded care for children under 18 years • Funding depends on the choice of GP (bulk-billing or not) and hospital (public or private)

Resource generation • Ondansetron in syrup (0.1 mg/kg) • Ondansetron in wafer ((8-15kg 2mg, 15-30kg 4mg, >30kg 6-8mg)

Service delivery • GP care always first

• Direct visits to emergency department are discouraged • Multiple options for seeking first care 

• Direct visits to emergency department are part of organization of daily care 

6. Re registration with a single provider, how does this ensure access when needed as well as after-hours care? Further explanation for audiences that are unfamiliar with the Dutch system would be helpful.

Thank you for the comment. We reformulated the sentence to ‘at the out-of-hours facilities, locum GPs are after working hours available for (telephone) consultations. Every citizen has access to this after-hours care.’ (Page 15 line 275-276). 

7. Re financing in Australia and PHI. My understanding is that PHI covers care provided as a private hospital patient (in a public or private hospital) as well as dental, psychological and allied health care but not medical services or medicines.

We mention the public health insurance in Australia on Page 11, line 188 (Citizens can choose to purchase extra private health insurance to access additional healthcare services (largely hospital care) and benefits not covered by Medicare.) It depends on the type of public health insurance how much is financed. As for the medication prescribed in general practice, they are typically issued as a private prescription and the caregivers are responsible for the costs (page 13, line 223). For the rotavirus vaccinations these are covered through the Public health services. 

8. The authors mention outbreak management as a key public health function of both systems. Is this funded differently?

Yes, both systems cover outbreak management via public health funds. However, we have now left out the outbreak management (question 4, reviewer 2) as we agree that public health is broader than outbreak management alone. 

9. The authors may like to consider using standard terminology for describing the funding arrangements for clinical services (eg fee for services, capitation etc) as this might be more readily understood by international audiences. Re funding in Australia, Medicare does not remunerate the GP, it reimburses the patient for services provided. This has not been adjusted for 10 + years resulting in increased co-payments and out of pocket costs for the patient.

Thank you for the suggestion. We adjusted the sentences for financing clinical care in The Netherlands to ‘The funding system for GPs involves a combination of fee-for-service and capitation-based reimbursement, along with some additional payments for specific services. The fee is based on the type of service and is subject to negotiation between healthcare providers and health insurers. The capitation fee is determined based on factors like the patient’s age, gender, and health status. Fees are adjusted annually to account for inflation and changes in the practice’s patient demographics. The Netherlands has a mandatory health insurance system, and individuals are required to have basic health insurance coverage. (Page 12, line 194-202). 

We have reformulated the sentences around Medicare as per your suggestions above: 

‘In Australia, the payment structure for GPs is primarily based on a fee-for-service model. GPs charge a fee for each service provided to patients. Patients typically pay the GP directly and then claim a rebate from Medicare. Medicare does not remunerate the GP but it reimburses the patient for services provided. However, the remuneration has not been adjusted for over ten years for inflation and rising costs, often leading GPs to charge higher fees, resulting in increased co-payments and out-of-pocket costs for patients. Some GPs offer ‘bulk-billing’, where Medicare covers the full consultation cost, and GPs bill Medicare directly instead of patients.’ (Page 12, line 209-216)

10. Why does differences in use of Ondansetron matter for your research question? Re the PBS, this does not cover paracetamol or Oral rehydration solutions (or ondansetron generally) except for ve

---

## [Decision Letter · Decision Letter 1]

28 May 2024

PONE-D-23-40570R1Comparing healthcare systems between the Netherlands and Australia in management for children with acute gastroenteritisPLOS ONE

Dear Dr. Weghorst,

Thank you for submitting your manuscript to PLOS ONE. After careful consideration, we feel that it has merit but does not fully meet PLOS ONE’s publication criteria as it currently stands. Therefore, we invite you to submit a revised version of the manuscript that addresses the points raised during the review process.

A rebuttal letter that responds to each point raised by the academic editor and reviewer(s). You should upload this letter as a separate file labeled 'Response to Reviewers'.• A marked-up copy of your manuscript that highlights changes made to the original version. You should upload this as a separate file labeled 'Revised Manuscript with Track Changes'.• An unmarked version of your revised paper without tracked changes. You should upload this as a separate file labeled 'Manuscript'.

We look forward to receiving your revised manuscript.

Kind regards,

Victor Daniel Miron

Academic Editor

PLOS ONE

Journal Requirements:

Reviewers' comments:

Reviewer's Responses to Questions

**Comments to the Author**

1. If the authors have adequately addressed your comments raised in a previous round of review and you feel that this manuscript is now acceptable for publication, you may indicate that here to bypass the “Comments to the Author” section, enter your conflict of interest statement in the “Confidential to Editor” section, and submit your "Accept" recommendation.

Reviewer #1: All comments have been addressed

Reviewer #2: (No Response)

2. Is the manuscript technically sound, and do the data support the conclusions?

Reviewer #1: Yes

Reviewer #2: Yes

3. Has the statistical analysis been performed appropriately and rigorously? 

Reviewer #1: N/A

Reviewer #2: N/A

4. Have the authors made all data underlying the findings in their manuscript fully available?

Reviewer #1: Yes

Reviewer #2: Yes

5. Is the manuscript presented in an intelligible fashion and written in standard English?

Reviewer #1: Yes

Reviewer #2: Yes

6. Review Comments to the Author

Reviewer #1: All my suggestions have been adequately addressed. The final manuscript reads well and my congratulations to the submitting authors.

Reviewer #2: The edits made by the authors has substantially improved the paper, and I commend the authors in their efforts. The aim of the paper and how this analysis sets out to address this is much clearer.

Some final minor points for attention.

I note the authors have mentioned rotavirus vaccination and specific therapies (eg ondansetron) without reference as to why these are relevant to childhood gastroenteritis. For unfamiliar readers, I would suggest briefly noting why it is relevant at first mention (this is described much later in the paper) or give a very brief overview of current best public health and clinical care practice for the prevention and management of childhood gastroenteritis in the introduction.

Table 1 is terrific and really helps with orienting the reader as to what is coming in the results. This needs a footnote to explain the ‘no jab, no pay’ policy.

The sentence ‘The methods of this study were based on a previous performed advisory report comparing health’ needs review for grammatical correctness. Also, with respect to this could the authors please briefly explain why this method was appropriate to this study aim, and a brief overview of what was involved.

In the methods and under study limitations, assuming no differences across jurisdictions is unreasonable; each will have differing policies, funding arrangements, delivery of services – albeit quite likely minor. This should be amended in the paper.

Re the sentence “Citizens can choose to purchase extra private health insurance to access additional healthcare services (largely hospital care) and benefits not covered by Medicare.” The reference to ‘and benefits’ is vague and adds little. I suggest deleting this.

I would suggest delete “, often leading GPs to charge higher fees’ from the following sentence for clarity, and ‘amend resulting in’ to ‘and has been associated with’ to avoid inferring causality.

“However, the remuneration has not been adjusted for over ten years for inflation and rising costs, often leading GPs to charge higher fees, resulting in increased co-payments and out-of-pocket costs for patients. Some GPs offer ‘bulk-billing’, where Medicare covers the full consultation cost, and GPs bill Medicare directly instead of patients.”

In addition, GP’s accept payment directly from Medicare on behalf of the patient, rather than billing Medicare directly. The text should be amended to reflect this.

Bulk-billing at first mention (whether the table or text) needs explanation for international audiences. So too in the abstract or use a different term more widely understood (eg. Without a co-payment).

In the sentence “Medication prescribed in general practice is typically issued as a private prescription, and the caregivers are responsible for the costs.” This needs to be qualified that this is medications for childhood gastroenteritis not medicines in general.

Re the sentence “For clinical care, the availability of medical equipment for the management of childhood gastroenteritis in primary care is minimal in both countries.” This still needs justification/clarification. Diagnostics and treatments are available in primary care for GE, but unlike in hospital settings, requires attending a second service provider. In addition, while point of care testing (presumably for electrolyte disturbances?) is generally not available in primary care, these are available in some remote/very remote primary care settings in Australia.

With respect to R1 Q3: I agree providing figures would help elucidate the point by giving a quantum reference for comparison (both for within each country as well between the countries).

In the sentence “The Australian system acknowledges the various ways in which unscheduled care may be accessed, providing caregivers with options beyond the traditional GP visit,” what is meant by ‘traditional’ is unclear. Please revise.

The paper should have general review for typographic and grammatical errors.

7. PLOS authors have the option to publish the peer review history of their article (what does this mean?). If published, this will include your full peer review and any attached files.

Reviewer #1: **Yes: **Michael Barrett

Reviewer #2: No

---

## [Author Response · Author response to Decision Letter 1]

13 Jun 2024

Response to Reviewers 2.0 

We would like to thank the editor for the opportunity to submit a revised version of our manuscript (PONE-D-23-40570). We thank the reviewers for the constructive comments and suggestions to improve our manuscript. The detailed response to each of their comments is provided in this document. We hope our revisions will meet the expectations put forward by the reviewers. The page and line numbers we refer to correspond to the ‘track changes’ document. 

Reviewer #1: All my suggestions have been adequately addressed. The final manuscript reads well and my congratulations to the submitting authors.

Thank you very much. 

Reviewer #2: The edits made by the authors has substantially improved the paper, and I commend the authors in their efforts. The aim of the paper and how this analysis sets out to address this is much clearer.

Some final minor points for attention.

1. I note the authors have mentioned rotavirus vaccination and specific therapies (eg ondansetron) without reference as to why these are relevant to childhood gastroenteritis. For unfamiliar readers, I would suggest briefly noting why it is relevant at first mention (this is described much later in the paper) or give a very brief overview of current best public health and clinical care practice for the prevention and management of childhood gastroenteritis in the introduction.

Thank you for the comment. We added the sentence: ‘Acute gastroenteritis is a highly contagious disease with rotavirus being the most common cause (2, 23)’ (Page 10, line 162). 

We added the word ‘an antiemetic medicine’ after ondansetron to give clarity (Page 16, line 303). 

In addition, we added it being an anti-emetic in the abstract. 

2. Table 1 is terrific and really helps with orienting the reader as to what is coming in the results. This needs a footnote to explain the ‘no jab, no pay’ policy.

Thank you for the compliment. We have added in the footnote of the table the following: ‘1No Jab No Pay: to access family assistance payments, immunisation is required. No Jab No Play: to attend childcare, immunisation is required.’ (Page 9, Table 1). In addition, because of the above comment, we added a footnote about ondansetron too. 

3. The sentence ‘The methods of this study were based on a previous performed advisory report comparing health’ needs review for grammatical correctness. Also, with respect to this could the authors please briefly explain why this method was appropriate to this study aim, and a brief overview of what was involved.

We have amended the sentence thus: The methods of this study were based on those used in a prior report comparing health systems between different countries (12). 

4. In the methods and under study limitations, assuming no differences across jurisdictions is unreasonable; each will have differing policies, funding arrangements, delivery of services – albeit quite likely minor. This should be amended in the paper.

Everything that is stated in the manuscript is applicable for each jurisdiction. We added ‘The primary care system and National immunisation program funding and policies are run by the Commonwealth and do not vary in application by jurisdiction.’ To the limitations (Page 21, line 401-403) 

5. Re the sentence “Citizens can choose to purchase extra private health insurance to access additional healthcare services (largely hospital care) and benefits not covered by Medicare.” The reference to ‘and benefits’ is vague and adds little. I suggest deleting this.

We deleted the ‘and benefits’ (Page 11, line 189). 

6. I would suggest delete “, often leading GPs to charge higher fees’ from the following sentence for clarity, and ‘amend resulting in’ to ‘and has been associated with’ to avoid inferring causality.

We deleted the ‘often leading GPs to charge higher fee’ (Page 12, line 213, 214) and reformulated ‘resulting in’ to ‘has been associated with’ (page 12, line 214). 

7. “However, the remuneration has not been adjusted for over ten years for inflation and rising costs, often leading GPs to charge higher fees, resulting in increased co-payments and out-of-pocket costs for patients. Some GPs offer ‘bulk-billing’, where Medicare covers the full consultation cost, and GPs bill Medicare directly instead of patients.”

In addition, GP’s accept payment directly from Medicare on behalf of the patient, rather than billing Medicare directly. The text should be amended to reflect this.

Thank you, the sentence has been amended thus:

Some GPs offer ‘bulk-billing’, a term which means they do not charge the patient a co-payment, rather Medicare covers the full consultation cost and patients authorise Medicare to pay the GP directly on their behalf.

8. Bulk-billing at first mention (whether the table or text) needs explanation for international audiences. So too in the abstract or use a different term more widely understood (eg. Without a co-payment).

We agree with the reviewer that it is difficult to explain bulk-billing as early as the abstract so have altered it to co-payment but we describe later in the manuscript the concept of bulk-billing as per the sentence above. 

9. In the sentence “Medication prescribed in general practice is typically issued as a private prescription, and the caregivers are responsible for the costs.” This needs to be qualified that this is medications for childhood gastroenteritis not medicines in general.

We added ‘for childhood gastroenteritis’ in the sentence (Page 13, line 223). 

10. Re the sentence “For clinical care, the availability of medical equipment for the management of childhood gastroenteritis in primary care is minimal in both countries.” This still needs justification/clarification. Diagnostics and treatments are available in primary care for GE, but unlike in hospital settings, requires attending a second service provider. In addition, while point of care testing (presumably for electrolyte disturbances?) is generally not available in primary care, these are available in some remote/very remote primary care settings in Australia.

We added ‘Primary care is not typically set up to monitor vital signs in an ongoing way or give fluids other than orally’ (Page 14, line 251-252) 

11. With respect to R1 Q3: I agree providing figures would help elucidate the point by giving a quantum reference for comparison (both for within each country as well between the countries).

As reviewer 1 is very happy with the revised manuscript we are not sure what reviewer 2 wishes us to change here and we are happy we satisfied the comments made by reviewer 1. 

12. In the sentence “The Australian system acknowledges the various ways in which unscheduled care may be accessed, providing caregivers with options beyond the traditional GP visit,” what is meant by ‘traditional’ is unclear. Please revise.

We have removed the word ‘traditional’. 

13. The paper should have general review for typographic and grammatical errors.

We regret typographic and grammatical errors. We re-read the manuscript and corrected were appropriate. We would like to notice that we have two native speakers as co-author who checked the manuscript.

---

## [Editor Report · Decision Letter 2]

24 Jun 2024

Comparing healthcare systems between the Netherlands and Australia in management for children with acute gastroenteritis

PONE-D-23-40570R2

Dear Dr. Weghorst,

We’re pleased to inform you that your manuscript has been judged scientifically suitable for publication and will be formally accepted for publication once it meets all outstanding technical requirements.

Kind regards,

Victor Daniel Miron

Academic Editor

PLOS ONE

---

## [Editor Report · Acceptance letter]

15 Jul 2024

PONE-D-23-40570R2 

PLOS ONE

Dear Dr. Weghorst, 

I'm pleased to inform you that your manuscript has been deemed suitable for publication in PLOS ONE. Congratulations! Your manuscript is now being handed over to our production team.

Kind regards, 

on behalf of

Dr. Victor Daniel Miron 

Academic Editor

PLOS ONE